# WaistonBelt X: A Belt-Type Wearable Device with Sensing and Intervention Toward Health Behavior Change

**DOI:** 10.3390/s19204600

**Published:** 2019-10-22

**Authors:** Yugo Nakamura, Yuki Matsuda, Yutaka Arakawa, Keiichi Yasumoto

**Affiliations:** 1Graduate School of Information Science, Nara Institute of Science and Technology, Nara 630-0192, Japan; 2Research Fellowship for Young Scientists of Japan Society for the Promotion of Science, Tokyo 102-0083, Japan; 3Graduate School of Science and Technology, Nara Institute of Science and Technology, Nara 630-0192, Japan; yasumoto@is.naist.jp; 4Graduate School and Faculty of Information Science and Electrical Engineering, Kyushu University, Fukuoka 819-0395, Japan; arakawa@ait.kyushu-u.ac.jp; 5JST Presto, Tokyo 102-0076, Japan

**Keywords:** wearable computing, mobile sensing, activity recognition, health behavior change, intervention, health care support system

## Abstract

Changing behavior related to improper lifestyle habits has attracted attention as a solution to prevent lifestyle diseases, such as diabetes, heart disease, arteriosclerosis, and stroke. To drive health behavior changes, wearable devices are needed, and they must not only provide accurate sensing and visualization functions but also effective intervention functions. In this paper, we propose a health support system, WaistonBelt X, that consists of a belt-type wearable device with sensing and intervention functions and a smartphone application. WaistonBelt X can automatically measure a waistline with a magnetometer that detects the movements of a blade installed in the buckle, and monitor the basic activities of daily living with inertial sensors. Furthermore, WaistonBelt X intervenes with the user to correct lifestyle habits by using a built-in vibrator. Through evaluation experiments, we confirmed that our proposed device achieves measurement of the circumference on the belt position (mean absolute error of 0.93 cm) and basic activity recognition (F1 score of 0.95) with high accuracy. In addition, we confirmed that the intervention via belt vibration effectively improves the sitting posture of the user.

## 1. Introduction

The ongoing epidemic of lifestyle diseases, including diabetes, heart disease, arteriosclerosis, and stroke, is regarded as an urgent issue in the world [1]. The main factors of these lifestyle diseases are unhealthy lifestyle habits, such as lack of exercise, wrong body posture, overeating, and snacking [2]. According to the report of McKinsey & Company, if the lifestyle habits of people do not improve, almost half of the adult population will be overweight or obese by 2030 [3]. Lifestyle diseases are preventable because they are primarily caused by the lifestyle habits of people. Therefore, we aim to solve this problem by promoting change from poor lifestyle habits to healthy behavior with wearable computing technology.

In recent years, many wearable devices and Internet of Things (IoT) systems for health care have been developed and released in the market [4,5,6]. These technologies mainly focus on remote health management based on sensing and visualization of the current status of users via smartphones or smartwatches. However, it is difficult to maintain and promote peoples’ motivation for health behavior change by relying only on the visualization of the user’s lifestyle using a smartphone or smartwatch application. To achieve long-term health behavior changes, not only visualization but also vibrotactile intervention during recognition of an accurate current state of the user in daily life are required. Therefore, how to design a health care support system to change peoples’ lifestyles using wearable and IoT technologies is still an unresolved and challenging issue.

Based on the above considerations, we have been developing a belt-type wearable device to measure peoples’ body and health conditions since 2014 [7,8]. The waist position is known to be the most suitable on-body position for user-independent activity recognition because the acceleration patterns are most similar for the same activity across different users [9]. In this paper, we propose a health support system WaistonBelt X (Figure 1) that provides sensing and intervention functions with a belt-type wearable device. In particular, our system aims to support health behavior change for preventing health problems due to improper lifestyle habits such as lack of exercise and wrong body posture. WaistonBelt X can automatically measure the waistline with a magnetometer that detects the movements of a blade installed in the buckle, and monitor basic activities of the user’s daily living with inertial sensors (3 axis accelerometer and gyroscope). WaistonBelt X also has a function for intervening in a user’s activity by vibrating. The vibrotactile intervention is triggered by a habit (activity) set by the user, such as a poor sitting posture. The measurement data are transmitted to our application via a Bluetooth low-energy (BLE) module. In our application, the stored data are visualized by graphs, and feedback from data analysis is given through a virtual assistant. Through evaluation experiments, we confirmed that our proposed device is able to measure the waistline (with a mean absolute error of 0.93 cm) and recognize basic activities of the user (F1 score of 0.95) with high accuracy in the 10-fold cross-validation. In the leave-one-person-out cross-validation, we confirmed that WaistonBelt X achieved an F1 score of 0.82. This recognition result is clearly accurate compared to when wearing sensors on the wrist which is the most standard position where the wearable device is worn. In addition, we confirmed that the intervention using the vibration on the belt works effectively to naturally improve the sitting posture of the user.

The contributions of this paper are summarized as follows:We have defined system requirements based on the concrete use case scenario, and designed a novel belt-type wearable device with a compact and stylish appearance named WaistonBelt X that allows monitoring the wearer’s activity all day and encouraging healthy behavior changesTo meet requirements, we have implemented WaistonBelt X which has the following functions: (a) Sensing functions (waistline measurement and recognition of the basic activities of daily living) and (b) intervention functions (context-aware belt vibration and short- and long-term visualization using a smartphone application).Through the in-the-wild experiments, we have confirmed that WaistonBelt X is able to measure the waistline with high accuracy, and recognize basic activities more accurately compared to wristband type wearable devices. Then, we have also confirmed that intervention functions with the vibrator of WaistonBelt X contribute to help naturally change poor habits such as bad sitting posture.

The paper is organized as follows: Section 2 describes our assumed use case scenarios and clarifies device requirements. Then, we describe our system, named WaistonBelt X, in Section 3 and evaluate the system in Section 4. Section 6 reviews related work and our previous work. Finally, Section 5 provides discussions of our study, and Section 7 concludes this paper.

## 2. Use Case and Requirements

In this section, we describe how health behavior change will be realized with our device, by introducing daily use case scenarios. Figure 2 shows four representative scenarios during one day of WaistonBelt X activity by user “Taro.”

### 2.1. Use Case Scenario

Taro (male, 36 years old) is an office worker in Japan. In the morning, Taro wakes up and starts to prepare for his job (Figure 2a). He wears the WaistonBelt X as usual, and it automatically measures his waistline. When he checks the smartphone application, today’s waistline is automatically recorded. After finishing the day’s preparations, he goes to his workplace. On the way to the workplace, WaistonBelt X monitors basic activities, such as how long he walked and whether he used stairs, with inertial sensors (Figure 2b). Arriving at the workplace, Taro starts working at his desk. WaistonBelt X recognizes when he starts to work and switches to posture monitoring mode. Taro has always suffered from a stiff neck, and he wants to correct his arched back. While Taro does desk work, WaistonBelt X detects that he is sitting with an arched back and intervenes to correct his sitting posture using several vibration patterns. By this intervention, Taro notices that his posture is wrong and can correct it (Figure 2c). WaistonBelt X continues to regularly intervene to improve Taro’s sitting posture throughout the workday. In the evening, Taro finishes his work and goes home. Before bedtime, Taro checks his activities for the day through the smartphone application. He can see whether that the number of poor lifestyle habit events is fewer than it was yesterday, and this can improve his motivation (Figure 2d).

### 2.2. Requirements

According to the use case scenarios described above, the following functionalities are required for WaistonBelt X:**(A)** **Sensing function.** To collect Taro’s physical state, such as waistline, measurement methods naturally integrated into his daily life should be provided. In addition, to monitor Taro’s lifestyle habits and intervene effectively with appropriate timing, a method to recognize basic activities of daily living such as walking, sitting, standing and condition of each activity such as the sitting posture is needed. Then, these sensing functions must be provided by a compact and wireless device.**(B)** **Intervention function.** To change Taro’s lifestyle habits, not only visualization but also vibrotactile intervention should be provided. This intervention should be based on the output of the sensing function. To improve Taro’s understanding of the current state of his body, visualization of the current waistline and posture is required. Additionally, based on the current state, WaistonBelt should intervene with Taro by using both the smartphone application and an actuator in the wearable device. To maintain Taro’s long-term motivation, a virtual agent (e.g., virtual assistant) who gives feedback and advice based on long-term lifestyle changes and trends should be provided.**(C)** **Battery lifetime, Privacy, and Security.** To monitor Taro’s activities all day, WaistonBelt X should continuously work as long as possible. Therefore, a mechanism for continuously monitoring while reducing processing with high power consumption such as transmitting raw data to a telephone by wireless communication is required. In addition, privacy and security are important for wearable devices since data about a user’s lifestyle habits are privacy-sensitive information. Data communication and storage of WaistonBelt X need to be secure.

## 3. WaistonBelt X

Considering the requirements summarized in the previous section, we propose a new belt-type wearable device named WaistonBelt X, as shown in Figure 1. To investigate the feasibility of WaistonBelt X, in this paper, we focused on requirements (A) and (B). The system consists of three parts: Wearable device, smartphone application, and cloud server. The wearable device measures the wearer’s body condition and lifestyle habits and directly intervenes by using an actuator in the device. The data measured by the wearable device is sent to the cloud server through the smartphone application, for analyzing the data. The smartphone application visualizes analyzed data and gives feedback to the wearer.

### 3.1. Hardware

WaistonBelt X is designed to attach to a common belt buckle. When placed on the buckle of a belt, the device can provide all of the needed functions. The device appearance is shown in Figure 1 and Figure 3. WaistonBelt X has inertial sensors (accelerometer, magnetometer, and gyroscope) to measure body condition and lifestyle habits, and an actuator (vibrator) to provide intervention. To communicate with the smartphone application, a BLE module (nRF52 from Nordic Semiconductor in Norway) is incorporated.

The main circuit of WaistonBelt X is implemented using a SenStick multi-sensor board, which we published previously [10]. The specs of this board allow operating about 1 h per 10 mAh battery. For example, WaistonBelt X will work about 11 h by using a 110 mAh battery. The operating time of WaistonBelt X depends on the battery capacity, and it is possible to increase the operating time by installing a larger capacity battery. SenStick is published as open-source hardware, including firmware and sample program, on Github (https://github.com/ubi-naist/SenStick).

### 3.2. Software

#### 3.2.1. Measurement of Waistline

WaistonBelt X measures the waistline using the belt position (the inner length of the circle formed by the belt, hereinafter referred to as waistline) by calculating the distance the belt is inserted into the buckle and the known total length of the belt. In our previous work, a rotary encoder was adopted to measure the insertion distance [7,8]; however, such mechanical components often cause the problems of increased device size and eventual hardware wear. To avoid mechanical sensors, we used the mechanism of a common belt, as shown in Figure 4a. The back surface of this belt was modified to have a sawtooth profile with peak-to-peak intervals of around 0.6 cm), and the buckle was modified to hold a pivoting metal blade. This blade moves up and down iteratively in synch with the insertion of the belt. WaistonBelt X measures this movement with its magnetometer, as shown in Figure 4b, and calculates insertion distance of the belt by counting peaks in the magnetometer time-series data.

#### 3.2.2. Daily Activity/Posture Rrecognition

WaistonBelt X recognizes basic activities of daily living using inertial sensors (3 axis accelerometer and gyroscope).

During data pre-processing, first a median filter and a 20-Hz third-order Butterworth low-pass filter are applied to 3 axis acceleration (Acc XYZ) and gyroscope (Gyro XYZ) signals to remove noise, such as spike noise. Because gravity and motion components are mixed in the denoised acceleration signal (Acc XYZ), gravity components (Gravity Acc XYZ) and motion components (Body Acc XYZ) signals are separated using a 0.3-Hz Butterworth low-pass filter. Furthermore, the magnitude signal (Gravity Acc Mag, Body Acc Mag, and Body Acc Jerk Mag) and the jerk signal (Body Acc Jerk XYZ) are obtained by calculating the Euclidean distance of the 3 axis signal and the time derivative of motion components acceleration signal, respectively. In addition to the angular velocity of the body movement component (Body Angular Speed XYZ), the angular acceleration (Body Angular Acc XYZ) and the magnitude signal (Body Angular Speed Mag and Body Angular Acc Mag) are generated via the same process. Furthermore, among the above waveforms, fast Fourier transform is applied to seven waveforms, except for Gravity Acc XYZ, Gravity Acc Mag, and Body Angular Speed XYZ, to generate signals mapped in the frequency domain. As a result, a total of 17 types of signals (10 time-domain and seven frequency-domain signals), shown in Table 1, are extracted.

In the feature extraction process, the 17 signals obtained during pre-processing are divided into 50% overlaps with a 1.28-s time window (128 samples). The time domain features and frequency domain features shown in Table 2 are calculated from signals separated by a time window of 1.28 s. As a result, 561 feature vectors are calculated from each data window. These features were selected based on previous research [9,11,12].

In the activity recognition model generation, 561 feature vectors are standardized and activity recognition models are generated based on a machine learning algorithm using standardized data as input. In this process, we use a random forest method, which was shown to be effective in previous research [9]. Scikit-learn [13], a machine learning library, was used for generating the activity recognition models. WaistonBelt X monitors the user’s basic daily activities using the activity recognition model generated by the above process.

Figure 5 shows the raw acceleration data signal waveform obtained from the WaistonBelt X when the wearer is walking and sitting. First, WaistonBelt X recognizes the context such as wearer is walking or sitting by the basic activity recognition process described above. Then, WaistonBelt X judges whether the wearer’ sitting posture is good or bad. The sitting posture is closely related to the angle around the waist. Users can prevent back pain, stiff shoulders and improve a fundamental metabolic capacity by keeping the pelvis at the correct angle. Therefore, WaistonBelt X achieves a method of estimating the user’s posture from the posture angle of the belt using the fact that the device is worn around the pelvis. Specifically, WaistonBelt X uses Equation Set (1) to estimate the three angles shown in the Figure 6 from the acceleration sensor values. Then, the good posture is determined according to the following two policies: (a) The left and right tilt is horizontal (Y axis rotation ψ≃0 degree: See Figure 7), (b) The front and rear tilt is almost vertical (Z axis rotation ϕ≃ 80-90 degree: See Figure 8).
(1)$$\theta =\tan^{-1}\left(\frac{Acc_{X}}{\sqrt{Acc_{Y}^{2}+Acc_{Z}^{2}}}\right),\psi =\tan^{-1}\left(\frac{Acc_{Y}}{\sqrt{Acc_{X}^{2}+Acc_{Z}^{2}}}\right),\phi =\tan^{-1}\left(\frac{\sqrt{Acc_{X}^{2}+Acc_{Y}^{2}}}{Acc_{Z}^{2}}\right)$$

The recognized activities and posture in the smartphone application are used as a trigger for intervention. For example, WaistonBelt X promotes the user to walk a little when the user is sitting/lying for a long time, and notify the user to correct the posture when the user is sitting in a bad posture.

#### 3.2.3. Visualization and Intervention

WaistonBelt X provides two intervention approaches for changing health behavior: (1) Smartphone application, and (2) vibrator on the belt.

The smartphone application has short- and long-term intervention and visualization functions, as shown in Figure 9. For short-term interventions, real-time feedback is provided based on analysis of sensor output, as shown in Figure 9a. For example, the application feeds back real-time measurements of the waistline when the user puts the belt on, and it advises them to improve their posture when they are sitting on a chair. For long-term intervention, visualization of achievement, as shown in Figure 9b, is provided. For example, time-series measured and analyzed data is shown as a graph for checking long-term lifestyle changes and trends, and the virtual assistant motivates people to change lifestyle habits.

The vibrator of the wearable device greatly contributes to amplification of the effect of short-term intervention. For example, when poor posture is detected, the device alerts the user both with vibration and a smartphone application. In daily life, many notifications from multiple applications are delivered to a smartphone or smartwatch. These multiple notifications sometimes become noise that interferes with intervention for health behavior change; in other words, people may not notice the intervention. WaistonBelt X can clearly intervene tactilely using several vibration patterns by the vibrotactile actuator on the belt.

To start interventions during daily activities, we use the results of activity recognition as triggers. For example, when WaistonBelt X detects the activity sitting, the system automatically shifts to posture monitoring mode and provides interventions to the wearer.

#### 3.2.4. Communication Sequences

Figure 10 shows communication sequences between WaistonBelt X, WaistonBelt app, and the user.

Regarding the short-term intervention, a real-time communication sequence is used as shown in Figure 10a. WaistonBelt X is turned on when the user wears the belt (a-I), and initialize sensors for starting monitoring (a-II). Then, WaistonBelt X discovers and connects with WaistonBelt app, and starts to stream sensor data via Bluetooth Low Energy (a-III). While receiving the streaming data, WaistonBelt app continually recognize the status of the user such as activity and posture (a-IV). In accordance with the recognized status, WaistonBelt app selects intervention method such as vibration via belt and notification via smartphone (a-V). If “vibration via belt” was selected, WaistonBelt app notifies flag to WaistonBelt X for operating vibrator.

Regarding the long-term intervention, an opportunistic communication sequence is used as shown in Figure 10b. Here, WaistonBelt X accumulates the sensing data in its own memory, and periodically transmits the accumulated data to the smartphone application. This allows the WaistonBelt X to save power consumed by communication. After turning on (b-I) and initializing (b-II), WaistonBelt X automatically and implicitly starts to record sensor data. Recorded data are accumulated device memory (b-III). When WaistonBelt app gets close to the belt opportunistically, the app discovers the belt and establishes a Bluetooth connection. WaistonBelt X selects data to send depending on the request, and sends data to the app (b-IV). Using sent data, WaistonBelt app recognizes the activities of the wearer, including sitting posture (b-V), and select and provide the intervention through notification and visualization, etc., (b-VI).

## 4. Evaluation

In this section, we evaluate each essential component of WaistonBelt X: Measurement of waistline, recognition of daily activity, and intervention for health behavior change.

### 4.1. Measurement of Waistline

#### 4.1.1. Experimental Settings

This section provides an evaluation of the waistline measurement method. Our proposed method is based on an estimation of the insertion distance of the belt into the buckle using a magnetometer; hence, we evaluate the estimation accuracy of the insertion distance. The insertion distance for the experiment was set as approximately 31 cm as a maximum (see Figure 4a). The number of trials was 100. As a performance metric, we used the mean absolute/relative error.

#### 4.1.2. Experimental Result

Table 3 shows the experimental result of belt insertion distance estimation using magnetometer. The average and standard deviation (SD) of absolute error of the estimation when the user inserts the belt approximately 31 cm into the buckle are 0.93 cm and 1.01 cm, respectively. Because the peak of the sawtooth shape on the belt is 0.6 cm (which corresponds to the resolution of our method), we confirmed that our proposed method performs with high accuracy. According to the mean relative error of 3.0%, our proposed method based on magnetometer has sufficient accuracy for use in daily life.

### 4.2. Daily Activity Recognition

#### 4.2.1. Experimental Settings

This section provides an evaluation of the daily activity recognition method. In order to evaluate the effectiveness of WaistonBelt X, we compare the recognition performance of different body positions such as not only the waist but also the wrist, chest, ankle. As target labels for basic activity recognition, we selected seven activities: (1) Lay down, (2) sit, (3) stand, (4) walk, (5) walk down stairs, (6) walk up stairs, and (7) running. We recruited 17 subjects (gender: 13 males and 4 females, age: 23.4 ± 1.0, height: 169.6 ± 6.3 cm, weight: 61.2 ± 10.3 kg). The weight and height of subjects are shown in the Figure 11. Each subject wear WaistonBelt X on waistline and SenStick on right wrist, left wrist, chest, right ankle, left ankle. Then, subjects conducted a data collection experiment where each subject carried out target activities on a university campus. As a result, a dataset for a total of about 250 h of activity was collected.

#### 4.2.2. Experimental Results

In the experiments, we considered two case validations: 10-fold cross-validation and one-person-out cross-validation.

In the 10-fold cross-validation, the dataset was randomized and divided into 10 groups with each group covering the same ratio of classes. In each fold, sensor data belonging to nine groups were used for training data and the remaining group’s sensor data was used for test data. In the leave-one-person-out cross-validation, in each fold, 16 persons were used for training data and the remaining person was used for test data.

Figure 12a shows the confusion matrix of basic activity recognition results for 10-fold cross-validation. The results clearly show that the algorithm for classification of basic activities performed very well (F1-score of 0.95). Figure 12b shows the confusion matrix of basic activity recognition results in leave-one-person-out cross-validation. The results show that the proposed method achieved an F1 score of 0.82, and some misclassifications occurred between sitting and standing, walking, and walking up and down stairs. We consider that the cause of accuracy was reduced because of individual differences, such as physique and gender, among subjects. In order to investigate the effect of the difference in physique between male and female, the recognition results aggregated by gender in leave-one-person-out cross-validation is shown in the Figure 13. We confirmed that results of male subjects: Figure 13a achieved an F1 score of 0.86 and results of female subjects: Figure 13b achieved an F1 score of 0.67. As shown in the Figure 11, we can confirm that the point of the female subject is far from the other point. In other words, this result indicates that female subjects have a larger physique difference than male subjects. We consider that this is the reason why female results are inferior to male results. From this knowledge, in the case where data of many subjects can be acquired, training the recognition model of the group that selected subject close to the physique is effective for achieving a more accurate recognition result. Figure 14 show performance comparison of each body position. This result indicates that the waist position achieves activity recognition with the highest accuracy compared to other body positions. In particular, it is clearly more accurate than the wrist, which is the most standard position where the wearable device is worn. Since the belt is an item worn by office workers on a daily basis, we expect the approach to make the belt smarter such as WaistonBelt X will contribute to the further development of wearable devices. In addition, a combination of WaistonBelt X and wristband type wearable devices can be expected to recognize a higher level of context such as wearer is eating and working on computers while sitting.

### 4.3. Body Vibrotactile Intervention Based on Sitting Posture Monitoring

#### 4.3.1. Experimental Settings

In this experiment, we focused on the behavior change scenario for a user’s posture improvement, as shown in Figure 15 in order to evaluate the effectiveness of body vibrotactile intervention. In this scenario, we considered that it is possible to improve the user’s posture by having the belt provide vibrotactile intervention as a negative reinforcement when the user has a poor posture and stopping the intervention when the user has a good posture. The sitting posture was judged from the attitude angle (see Figure 6) of the WaistonBelt X. Specifically, we defined the state where the attitude angle was kept horizontall (Y axis rotation ψ≃0 degree) and vertical (Z axis rotation ϕ≃ 80-90 degree) as a good sitting posture. We conducted an experiment targeting 11 general users (males in their twenties) for the purpose of evaluating what kind of influence the vibrotactile intervention has on the posture of the user. During the experiment, the user wore the belt and did a total of 40 min of personal computer (PC) work. In addition, the effectiveness of the intervention was verified by switching between the presence or absence of the intervention at intervals of 10 min, as shown below.
Experiment 1: No intervention (10 min of PC work)Experiment 2: Intervention for posture improvement (10 min of PC work)Experiment 3: No intervention (10 min of PC work)Experiment 4: Intervention for posture improvement (10 min of PC work)

#### 4.3.2. Experimental Result

Figure 16 shows the results of the intervention experiment to improve posture. The horizontal axis indicates each user and the vertical axis shows the poor posture rate (proportion of time counted as poor posture during 10 min) from four experiments for each user. The results show that the poor posture rate decreases when intervention is performed. In addition, the overall posture improvement rate (corresponding to a reduction of poor posture) between experiments 1 and 2 by belt vibrotactile intervention was 84.8% on average. However, there are differences in the effectiveness of the vibrotactile intervention depending on the user’s attributes, as there are users with an increase or decrease in the poor posture rate between experiments 2 and 3 and 3 and 4.

Figure 17a shows the result of a questionnaire about the noticeability of intervention, specifically whether vibrotactile intervention from the belt is easy to notice in comparison with other notification devices (such as smartphones and smartwatch).

Figure 17b shows the result of a questionnaire about the discomfort when wearing WaistonBelt X. The result shows that by putting the core module (sensing and intervention mechanism) of WaistonBelt X into a buckle-size device, it is possible to educe the discomfort felt by the user.

## 5. Discussion

### 5.1. Sensing Function

In this paper, in order to confirm the feasibility of the concept of the sensorized belt, we have proposed the activity recognition model targeting basic activities in daily life. Through the evaluations, we have clarified the accuracy of activity recognition decreases depending on attributes of the wearer, such as gender. To solve this issue, introducing a personalized recognition model can be considered. For instance, a common model is used at the beginning, and it will be optimized for each individual by using online learning, etc.

In addition, to provide more flexible intervention, WaistonBelt X should be able to recognize more context. Hence, we will investigate a combination of WaistonBelt X and other IoT/wearable sensors that aim to achieve a higher level of context recognition such as wearer is eating and working on computers while sitting.

### 5.2. Intervention Function

Regarding short-term intervention, since it is difficult to cover all intervention scenarios, we evaluated intervention function while focusing on sitting posture as a case study. In the future, we will conduct a large-scale and long-term intervention experiment that takes multiple scenarios into consideration and investigate the effect of differences in intervention methods and timing. Then, we will implement a mechanism for selecting the most appropriate means from among multiple intervention methods.

Regarding long-term intervention, we implemented a visualization function for measured and analyzed data. To intervene effectively, visualization methods and notification mechanisms that make a habit that the user regularly checks his/her physical status will become future work.

### 5.3. Battery Lifetime, Privacy, and Security

WaistonBelt X aims to monitor the wearer all day. The current implementation of WaistonBelt X can operate about 11 h with a 110 mAh battery on the specification of the mainboard, but a mechanism to reduce power consumption is essential considering actual usage. Therefore, methods to achieve low energy such as exploiting energy harvesting elements [14,15] and hardware-friendly activity recognition model [16,17] become promising solutions. Furthermore, it is possible to extend the operating time by installing the latest low power consumption controller chip.

Privacy and security are important issues for WaistonBelt X since data about a user’s lifestyle habits are privacy-sensitive information. In the future, we need to extend the belt by integrating privacy-aware data management methods [18,19] and secure authentication protocols [20]. In addition, how to design an open platform for encouraging large scale data collecting and knowledge sharing based on previous research [21,22] is also an important aspect for widespread adoption.

## 6. Related Work and Previous Work

### 6.1. Related Work

#### 6.1.1. Wearable Devices Focusing on Health Care

Thanks to the popularity of wearable devices in the world, we can collect biological data and momentum data easily on a daily basis. Many kinds of wearable devices have already been utilized.

Withings produces wrist-watch-type wearable devices [23], including Withings Active and Withing Pulse O${}_{2}$. These devices measure biological and momentum data and upload them to the cloud. Furthermore, this system allows management and analysis of the data, which can be combined with data measured by other health care devices. JINS MEME [24] is a wearable eyewear device for examining eye movements using 3-point electrooculography sensors and tracking very small changes in a user’s body using six axis sensors. It captures changes in the user’s blinks and eye movements to determine the user’s physical condition, including the level of concentration, fatigue, or composure.

#### 6.1.2. Activity Recognition Using Wearable Devices

Human activity recognition using wearable devices has been an active research topic for more than a decade in the ubiquitous computing field [25,26]. Recognizing the daily activities of users, such as sitting, laying down, walking, or running, can be useful information for understanding one’s own lifestyle. Many existing methods are focused on user activity recognition using inertial sensors, such as accelerometers, gyroscopes, and magnetometers, in wearable devices. Yang et al. [27] used accelerometers to identify users’ activities, such as standing, walking, running, and PC use. Du et al. [28,29] recognizes four levels of group mobility, including stationary, strolling, walking, and running, based on accelerometers and magnetometers. Sztyler et al. [9,30] proposed a robust activity recognition system based on on-body localization using wearable accelerometers. Moreover, they revealed that waist position is the most suitable place among all on-body positions (head, upper arm, forearm, chest, waist, thigh, and shin ) for user-independent activity recognition. The reason is because the acceleration patterns at the waist are most similar for the same activity across different users. Due to this fact, we focused on the belt, which is a daily object attached to the waist position, and developed a belt-type wearable device.

#### 6.1.3. Intervention Using Wearable Devices

Continuous motivation is required for maintaining human health. Many large technology companies produce wearable devices associated with health care services [31,32,33]. However, mainstream functionality of such services is recording and reporting information about behaviors such as physical activity or sleep patterns. These services do not provide a straightforward intervention to the user, so it is still difficult to bridge the gap between information gathering and behavioral change [34]. To improve the motivation of a user for changing behavior, effective intervention with a smart device is essential. Some intervention systems using wearable vibrotactile actuators have been proposed for the purpose of route navigating for a user [35]. Linda et al. [36] showed that vibrotactile navigation systems can be used in violent outdoor environments and are more useful than visual displays under conditions of high cognitive and visual workload. Tsukada et al. [37] and Jan et al. [38] proposed a navigation system using a belt-type vibro-tactile device. In addition, they showed that the front of the waistline is more sensitive to touch than in the back of the waistline. Based on the above factors, we embedded a vibrator at the belt buckle position and adopted the vibrotactile stimulation for intervention to change health behavior.

#### 6.1.4. State-of-the-Art of Belt-Type Wearable Devices

To change poor lifestyle habits, users should measure objective data, such as waistline, overeating, activity, and posture [39]. There are several companies and studies that have proposed a belt-type wearable device to achieve this.

Much effort has been expended to manage daily activities. Belty [40], which was announced at CES2016, is a belt-type wearable device with the following functions: activity tracker, step counting, and artificial intelligence for advising the user. Belty measures and analyzes the health data of users and provides advice based on the analyzed data. Additionally, the developer claims that it can encourage user motivation for changing lifestyle habits because it can give advice using artificial intelligence. However, because it can only give advice based on the user’s activities, the user cannot experience a sense of achievement obtained by the improvement of lifestyle habits. Another product, WELT [41], also provides activity tracking, measurement of waistlines, and eating habit estimation. However, it can only support the visualization of measured data and does not provide any method to improve lifestyle habits of the user.

To monitor posture, Hyejeong [42] proposed a posture detection method using force sensing and an accelerometer on a belt-attached device. An evaluation showed that this system could judge a user’s posture with a high accuracy of 88%. However, this proposed method can only recognize three types of posture: standing, sitting upright, and sitting bent. Moreover, because this evaluation did not consider the user’s body shape, it would be difficult to apply the device in real-world situations.

### 6.2. Previous Work

In our previous work, we developed the two prototypes shown in Figure 18.

The first prototype, named WaistonBelt [7] was shaped like a belt buckle. We proposed a method for measuring the waistline when the user wears the belt by embedding a sensor in the buckle. The mechanism for measuring the waistline was based on a rotary encoder, a mechanical sensor that measures differences of rotation angles.

The second prototype, named WaistonBelt 2 [8] was designed to update the WaistonBelt. The size of the device is 45% smaller compared with WaistonBelt, even though it had the same waistline measurement mechanism. Moreover, it had a pressure sensor for measuring the tension of the belt while it was worn it to detect overeating.

These two prototypes revealed the limitations of a mechanical architecture for sustainable sensing. For example, frictional wear causes a slip between the rotary encoder and the belt, which can decrease the measurement accuracy as time goes on. Hence, in this study, we employed a new architecture for measuring waistline based on magnetic field change when the belt is won. By this redesign, we have realized WaistonBelt X as an attachment to an existing belt buckle. Moreover, we have introduced two new functions (activity recognition and intervention) into WaistonBelt X for assessing the daily activities of the user, and improving lifestyle habits in daily life by giving feedback directly. Overall, WaistonBelt X can support health behavior change more efficiently.

## 7. Conclusions

The epidemic of lifestyle diseases, including diabetes, heart disease, arteriosclerosis, and stroke, is regarded as an urgent issue in the world. Lifestyle diseases are preventable because they are primarily based on the lifestyle habits of people. Therefore, we aim to solve this problem by promoting health behavior change, such as modifying poor lifestyle habits using wearable computing technology.

In this paper, we proposed the health support system WaistonBelt X (Figure 1), which provides sensing and intervention functions with a device worn on the belt. WaistonBelt X can automatically measure the waistline with a magnetometer and monitor basic activities of daily living using an accelerometer and gyroscope. WaistonBelt X also has a function for intervening with the user by vibrating. The measured data are visualized on graphs, and feedback is given through a virtual assistant with the analyzed data. Through evaluation experiments, we confirmed that our proposed device achieves measurement of the waist circumference using the belt position (the mean absolute error is 0.93 cm) and basic activity recognition (F1 score of 0.95) with high accuracy. In addition, we confirmed that vibrotactile intervention works effectively for improving the sitting posture of the user.

In the future, we will conduct long-term and large-scale fieldwork using WaistonBelt X to investigate the new insights it provides into the relationship between physical and mental health. Finally, we aim to develop novel "digital medicine" with wearable devices that will greatly contribute to physical and mental health maintenance and enhancement.

## Figures and Tables

**Figure 1 sensors-19-04600-f001:**
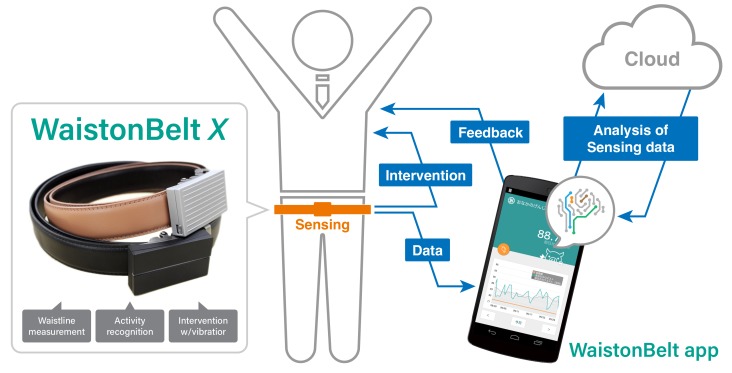
WaistonBelt X.

**Figure 2 sensors-19-04600-f002:**
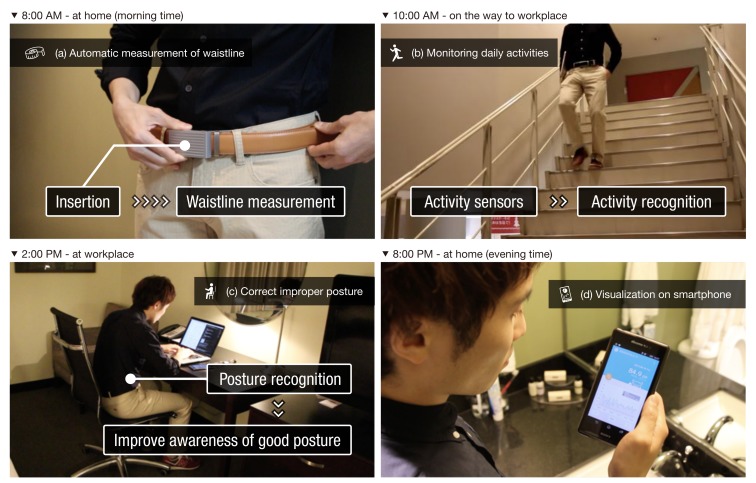
Use case scenario of WaistonBelt X.

**Figure 3 sensors-19-04600-f003:**
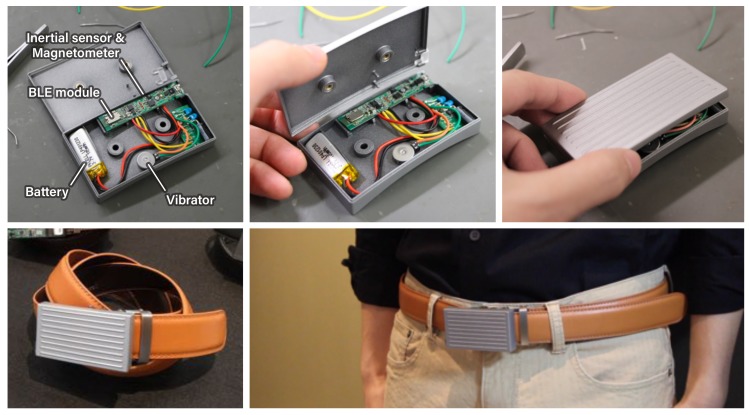
Device and circuit appearance.

**Figure 4 sensors-19-04600-f004:**
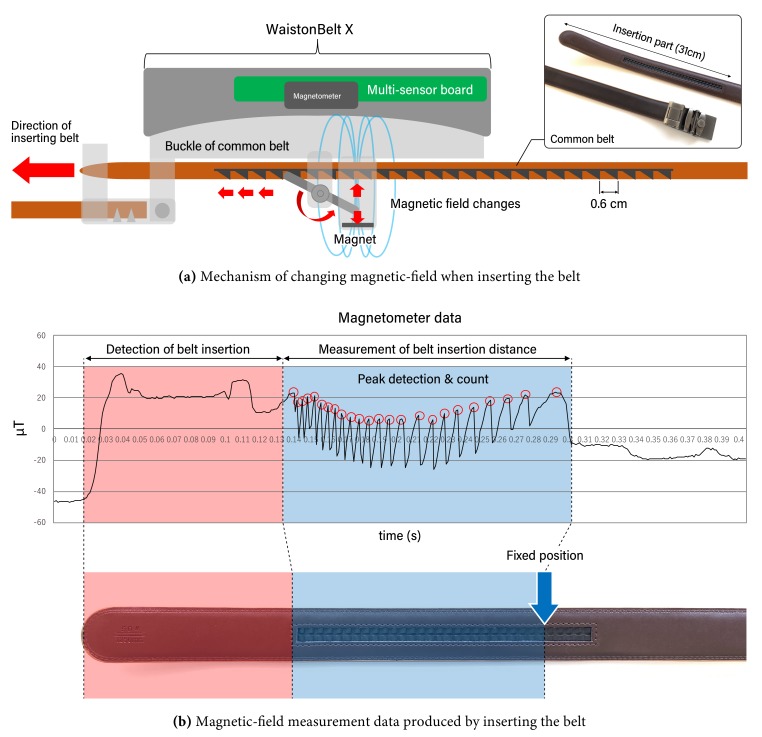
Mechanism of waistline measurement.

**Figure 5 sensors-19-04600-f005:**
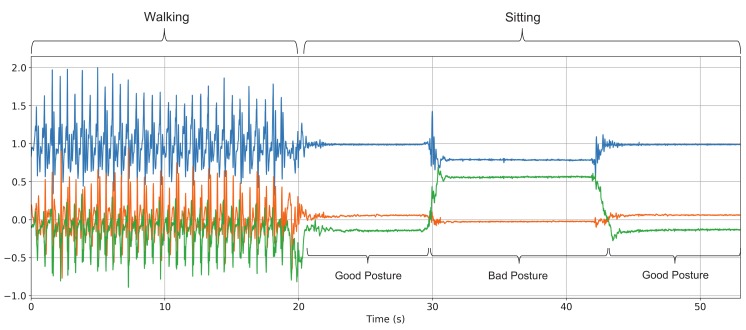
Raw acceleration data waveform obtained from the WaistonBelt X during walking and sitting.

**Figure 6 sensors-19-04600-f006:**
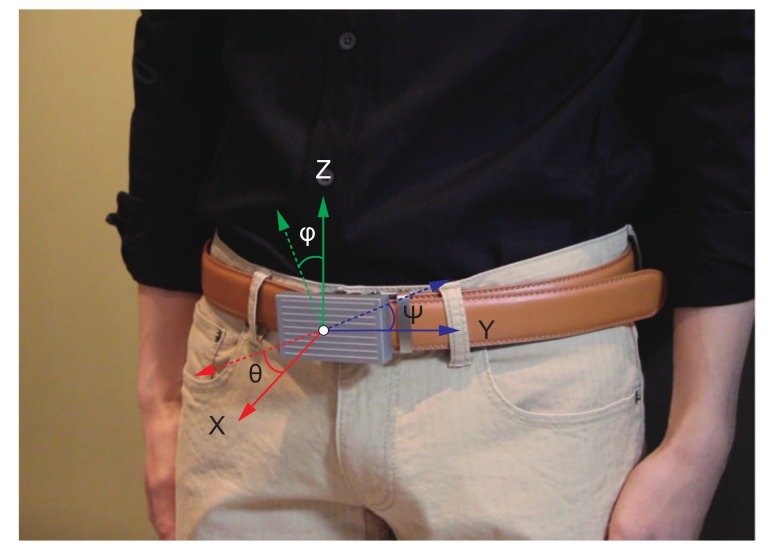
Attitude angle of WaistonBelt X.

**Figure 7 sensors-19-04600-f007:**
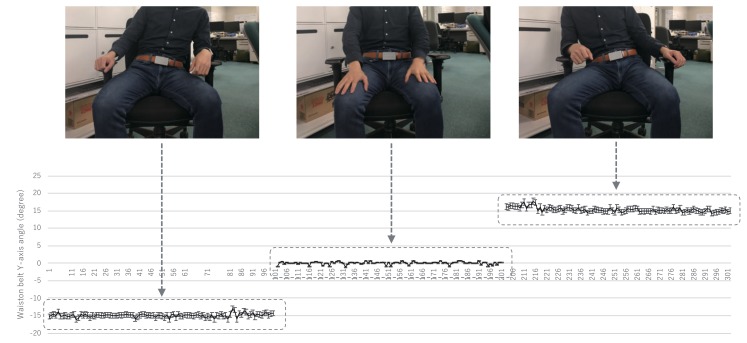
Relationship between Y axis angle and sitting posture.

**Figure 8 sensors-19-04600-f008:**
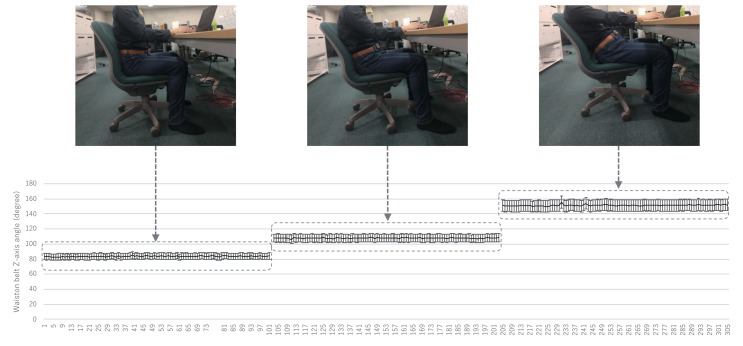
Relationship between Z axis angle and sitting posture.

**Figure 9 sensors-19-04600-f009:**
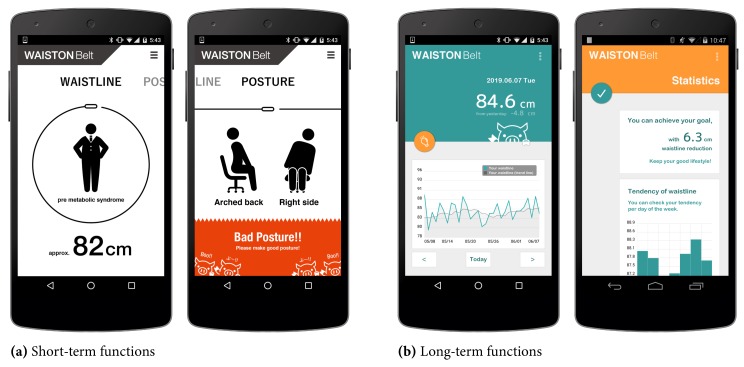
Smartphone application for short- and long-term intervention and visualization.

**Figure 10 sensors-19-04600-f010:**
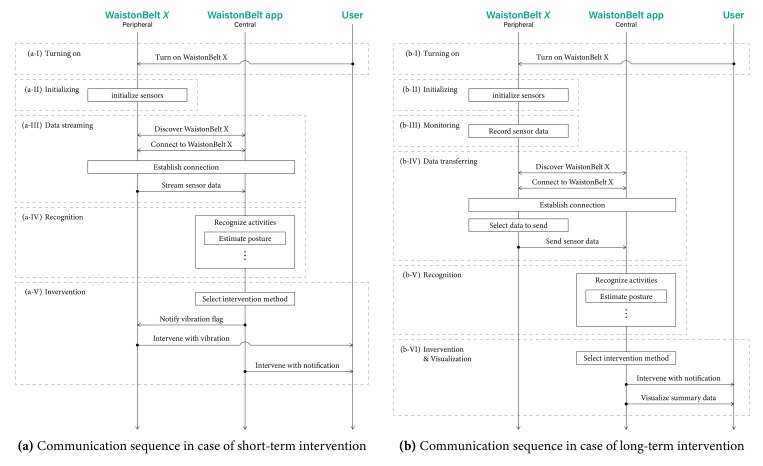
Communication sequence between WaistonBelt X, WaistonBelt app, and user.

**Figure 11 sensors-19-04600-f011:**
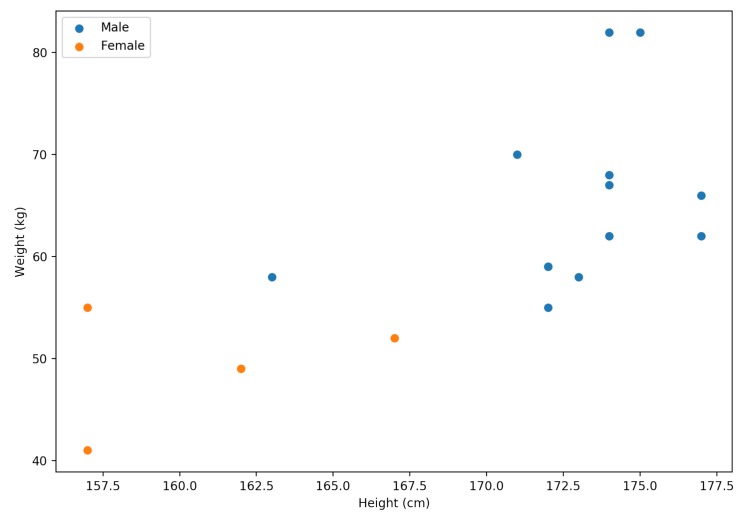
Weight and height of subjects.

**Figure 12 sensors-19-04600-f012:**
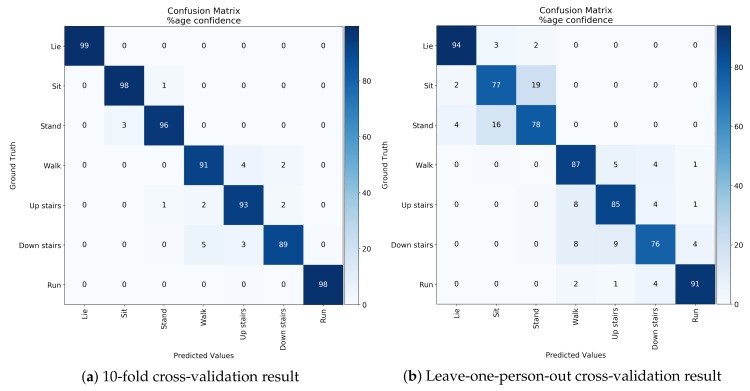
Result of cross-validation for basic activity recognition.

**Figure 13 sensors-19-04600-f013:**
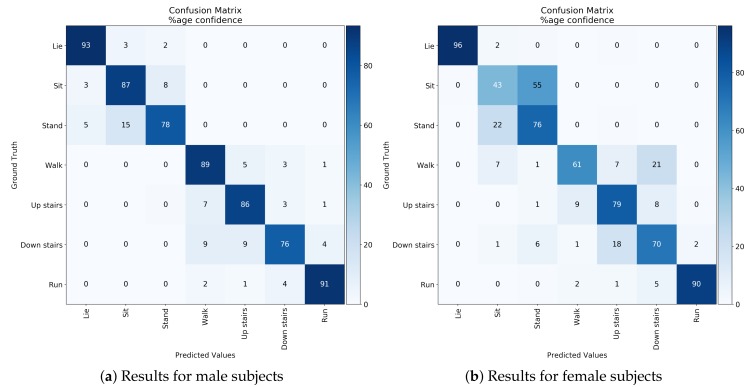
Effects of gender differences on recognition results (leave-one-person-out cross-validation).

**Figure 14 sensors-19-04600-f014:**
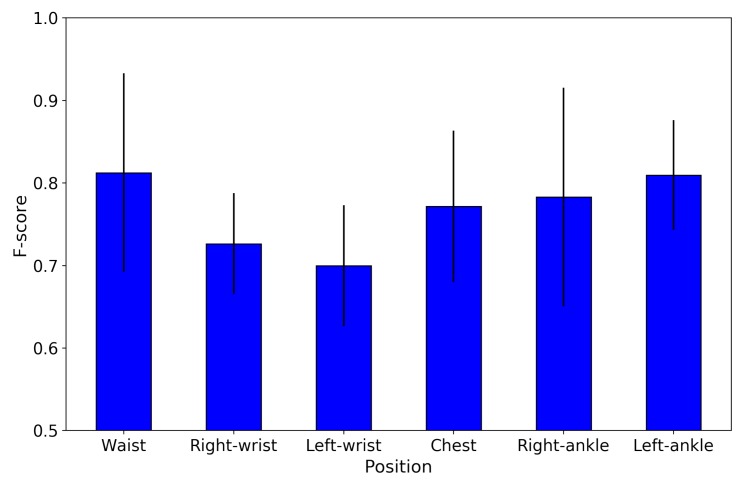
Difference in accuracy for each body position.

**Figure 15 sensors-19-04600-f015:**
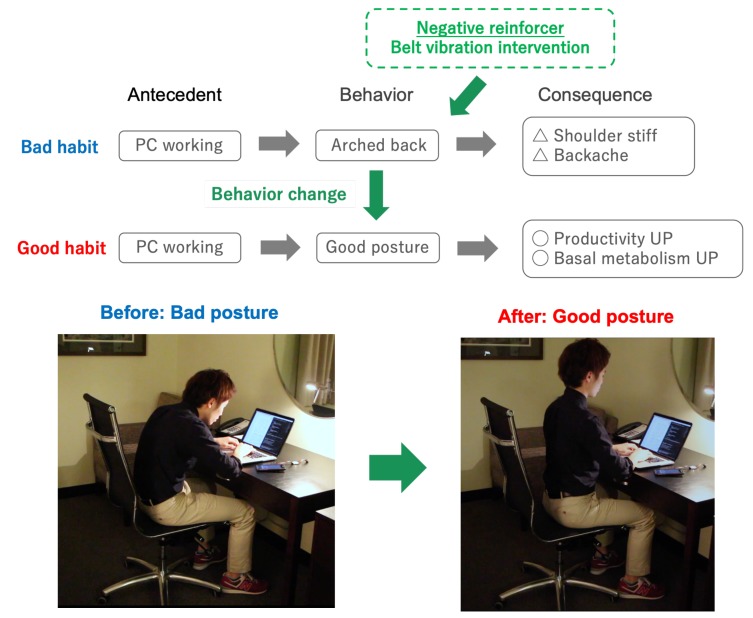
Assumed behavior change scenario (sitting posture improvement).

**Figure 16 sensors-19-04600-f016:**
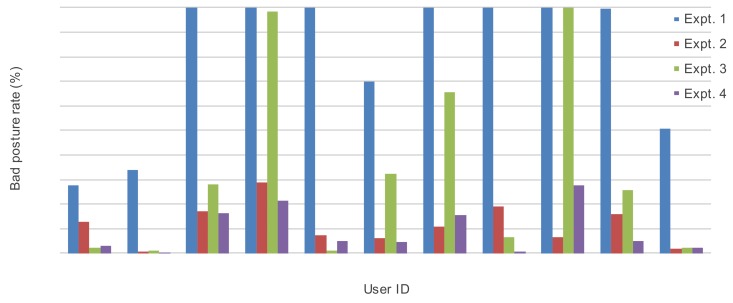
Experimental results of intervention to improve sitting posture.

**Figure 17 sensors-19-04600-f017:**
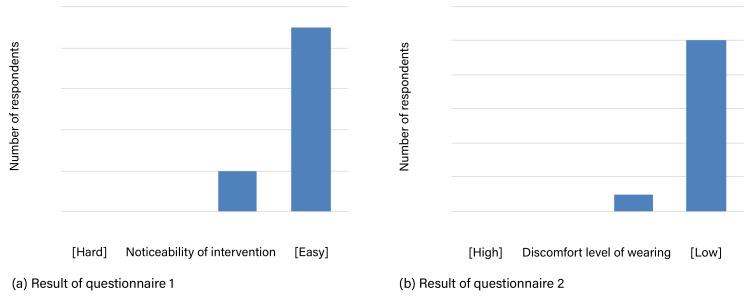
Results of questionnaires on intervention function.

**Figure 18 sensors-19-04600-f018:**
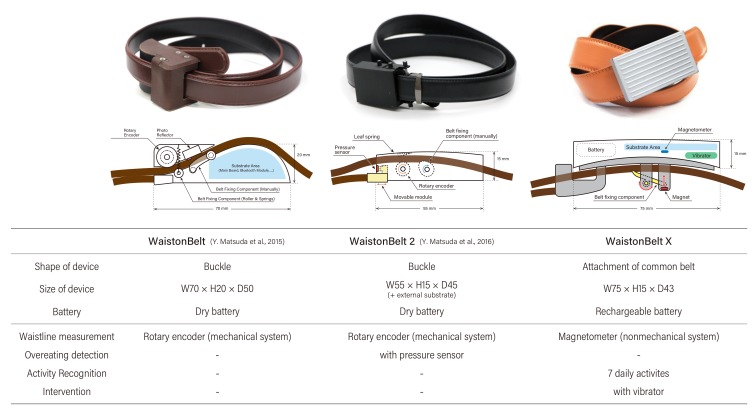
Previous work related to WaistonBelt.

**Table 1 sensors-19-04600-t001:** Time and frequency domain signals generated from accelerometer and gyroscope.

Signal Name (Acc: Acceleration, Mag: Magnitude)	Type (T: Time, F: Freq.)
Body Acc XYZ	T, F
Gravity Acc XYZ	T
Body Acc Jerk XYZ	T, F
Body Angular Speed XYZ	T, F
Body Angular Acc XYZ	T
Body Acc Mag	T, F
Gravity Acc Mag	T
Body Acc Jerk Mag	T, F
Body Angular Speed Mag	T, F
Body Angular Acc Mag	T, F

**Table 2 sensors-19-04600-t002:** Function list for feature extraction.

Function	Description	Formula	Type
mean(s)	Arithmetic mean	$\bar{s}=\frac{1}{N}\sum_{i=1}^{N}s_{i}$	T, F
std(s)	Standard deviation	$\sigma =\sqrt{\frac{1}{N}\sum_{i=1}^{N}{\left(s_{i}-\bar{s}\right)}^{2}}$	T, F
mad(s)	Median absolute deviation	$median_{i}\left(|s_{i}-median_{j}\left(s_{j}\right)|\right)$	T, F
max(s)	Largest values in array	$max_{i}\left(s_{i}\right)$	T, F
min(s)	Smallest value in array	$min_{i}\left(s_{i}\right)$	T, F
energy(s)	Average sum of the square	$\frac{1}{N}\sum_{i=1}^{N}s_{i}^{2}$	T, F
sma($s_{1}$, $s_{2}$, $s_{3}$)	Signal magnitude area	$\frac{1}{3}\sum_{i=1}^{3}\sum_{j=1}^{N}\left|s_{i,j}\right|$	T, F
entropy(s)	Signal entropy	$\sum_{i=1}^{N}\left(c_{i}\log\left(c_{i}\right)\right),c_{i}=s_{i}/\sum_{j=1}^{N}s_{j}$	T, F
iqr(s)	Interquartile range	Q3(s)−Q1(s)	T, F
autoregresion(s)	4th order Burg autoregression coefficients	$a=arburg\left(s,4\right),a\in R^{4}$	T
correlation($s_{1}$, $s_{2}$)	Pearson Correlation coefficient	$C_{1,2}/\sqrt{C_{1,1}C_{2,2}},C=cov\left(s_{1},s_{2}\right)$	T
angle($s_{1}$, $s_{2}$, $s_{3}$, v)	Angle between signal mean and vector	$\tan^{-1}\left(\right\|\left[{\bar{s}}_{1},{\bar{s}}_{2},{\bar{s}}_{3}\right]\times \upsilon \left\|,\left[{\bar{s}}_{1},{\bar{s}}_{2},{\bar{s}}_{3}\right]\cdot \upsilon \right)$	T
range(s)	Range of smallest value and largest value	$max_{i}\left(s_{i}\right)-mix_{i}\left(s_{i}\right)$	T
rms(s)	Root square means	$\sqrt{\frac{1}{N}\left(s_{1}^{2}+s_{2}^{2}+\cdot \cdot \cdot +s_{N}^{2}\right)}$	T
skewness(s)	Frequency signal skewness	$E[{\left(\frac{s-\bar{s}}{\sigma }\right)}^{3}]$	F
kurtosis(s)	Frequency signal kurtosis	$E\left[{\left(s-\bar{s}\right)}^{4}\right]/E{\left[{\left(s-\bar{s}\right)}^{2}\right]}^{2}$	F
maxFreqInd(s)	Largest frequency component	$argmax_{i}\left(s_{i}\right)$	F
meanFreq(s)	Frequency signal weighted average	$\sum_{i=1}^{N}\left(is_{i}\right)/\sum_{j=1}^{N}s_{j}$	F
energyBand(s, a, b)	Spectral energy of a frequency band [a, b]	$\frac{1}{a-b+1}\sum_{i=a}^{b}s_{i}^{2}$	F

N: Signal vector length, Q: Quartile, T: Time domain features, F: Frequency domain features, s: Sensor data divided for each time window (128 samples).

**Table 3 sensors-19-04600-t003:** Experimental result of belt insertion distance estimation.

Absolute Error [cm]			Relative Error [%]	
Average	Standard Deviation		Average	Standard Deviation
0.93	1.01		3.00	3.24

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
