# Peer review of "WaistonBelt X: A Belt-Type Wearable Device with Sensing and Intervention Toward Health Behavior Change"

_sensors, 2019, doi:10.3390/s19204600_

Round 1

Reviewer 1 Report

The authors developed a belt-type wearable sensing device by using a magnetometer, an accelerometer, and a gyroscope. The potential application was demonstrated and analyzed carefully, which confirmed the value of the developed system. Some suggestions are presented below.

1. As this is a scientific journal, the authors should summarize the scientific contribution of this work.

2. To clarify the proposed concept clearly, Figures 1 and 3 should be put together.

3. Figures 5, 6 and 7 should also be put together so that it would not make them look messy.

4. The detailed circuit diagram and the logic mechanism of the APP should be provided to reproduce this device by other researchers.

5. The real-time signals curves from the sensing device should be provided and how do the authors analyze the signals to distinguish and confirm different postures in Figures 10 and 11.

6. What are the limitations of this work and how to avoid and/or improve them?

Reviewer 2 Report

This paper presents an interesting and useful belt-type wearable device that aims to monitor user behavior and stimulate positive change. The proposed solution incorporates a wearable hardware component plugged to a belt, activity analysis software, and a smartphone application for data visualization and user interaction. The proposed solution also provides short-term intervention using vibration and long-term intervention by presenting data to the user through the smartphone app.

Although the paper is written clearly, a couple of critical points are either missing or unclear.

The proposed technique aims to detect (and intervene) bad body condition and poor lifestyle habits. However, neither of these conditions are defined formally.  

One of the examples in the paper is notifying the user when his/her posture is bad. But it is not clear how posture recognition is implemented. More importantly, it is not clear how a posture is classified as good or bad. Similarly, the authors do not explain how they conclude that a given life style is bad.

- Similar to the previous point, the user studies show that the proposed technique can differentiate different activities, such as sit and stairs up/down. However, the experiments do not evaluate the accuracy of detecting a bad posture, or in general, it is not clear how a poor body condition and poor lifestyle is recognized.

Without this clarification and results, it is not clear whether the stated goals are achieved.

- Is the sensor data processed locally on the wearable device or is it processed on the smartphone? From the hardware description, it seems that the sensor data is transmitted to the smartphone and the analysis software runs on the phone. If this is the case, does the phone send a signal back to the device to trigger vibration?

- The requirements (Section 2.2) miss two very important considerations: Battery lifetime/low energy consumption and privacy/security concerns.

1- Battery lifetime, hence, energy consumption, are vital for the adaptation of wearable devices. However, energy consumption requirement is not discussed at all. For example, low-power local processing is preferred over transmitting raw data to a phone since wireless communication is very costly. Even if this is not the main contribution of this paper, the low energy requirements and low-energy solutions to human activity recognition, such as the one below, need to be discussed as related work:

Bhat, et al. “Online human activity recognition using low-power wearable devices,” In Proc. Intl. Conference on Computer-Aided Design, Nov. 2018.

Anguita, Davide, et al. "Human activity recognition on smartphones using a multiclass hardware-friendly support vector machine." International workshop on ambient assisted living. Springer, Berlin, Heidelberg, 2012.

2- Similarly, privacy and security are important since they may prevent users from adopting wearable solutions. Are data communication and storage secure?

J. Espay et al., “Technology in Parkinson’s Disease: Challenges and Opportunities,” Movement Disorders, vol. 31, no. 9, pp. 1272–1282, 2016.

Bhat G, Deb R, Ogras UY. Openhealth: Open source platform for wearable health monitoring. IEEE Design & Test. March 2019.

It is reasonable that the authors do not address all requirements in one paper. However, they can mention these requirements and state that they focus only on sensing and intervention functionalities.

- The reasons for the misclassifications in Figure 9b can be discussed with more detail. 

Reviewer 3 Report

Authors have chosen a really nice problem which has clearly a significant societal impact. But proposed solution lacks the required technical details and experiment discussions.

Strengths

Problem is very relevant. introduction section excites the reader about the problem. Paper writing is lucid and it was easy to understand. Activity detection and intervention combined looks very promising

Weakness

   1. In introduction section,  Authors mention key reasons for unhealthy lifestyles are snacking, overeating, lack of exercise but they never discussed how to solve these issues anywhere in the paper. In stead they only focused on bad posture which is fine but they should have mentioned that other examples are not in the scope of this work.

2.  There is no technical or experimental discussion about how they differentiate between poor posture and good postures.

3. Selected seven activities are very basic and are studied extensively from past 10 years.

4. This work seems as an engineering work then research. It has good application value but limited scientific innovation.

Round 2

Reviewer 1 Report

Since most of the suggestions have been revised, I would like to recommend accepting this work.

Reviewer 2 Report

The revision addresses the recommendations made for the original submission. 

There are some minor spelling errors (Line 39, 300, 176, 177). 

Reviewer 3 Report

Authors have included all the changes requested.